# Effect of Drying Methods on Chemical Profile of Chamomile (*Matricaria chamomilla* L.) Flowers

Teuta Benković-Lačić [1,*] , Iva Orehovec [2] , Krunoslav Mirosavljević [1] , Robert Benković [1] ,
Sanja Ćavar Zeljković [3,4] , Nikola Štefelová [3] , Petr Tarkowski [3,4] and Branka Salopek-Sondi [2]

[1] Biotechnical Department, University of Slavonski Brod, 35000 Slavonski Brod, Croatia;
    kmirosavljevic@unisb.hr (K.M.); rbenkovic@unisb.hr (R.B.)
[2] Department for Molecular Biology, Ruđer Bošković Institute, 10000 Zagreb, Croatia;
    iva.orehovec@irb.hr (I.O.); branka.salopek.sondi@irb.hr (B.S.-S.)
[3] Czech Advanced Technology and Research Institute, Palacky University, 78371 Olomouc, Czech Republic;
    sanja.cavar@upol.cz (S.Ć.Z.); nikola.stefelova@upol.cz (N.Š.); petr.tarkowski@upol.cz (P.T.)
[4] Centre of the Region Haná for Biotechnological and Agricultural Research, Department of Genetic Resources
    for Vegetables, Medicinal and Special Plants, Crop Research Institute, 78371 Olomouc, Czech Republic
*   Correspondence: tblacic@unisb.hr

**Abstract:** Chamomile (*Matricaria chamomilla* L.) is used in the food industry, stomatology, pharmacy, and medicine due to the beneficial properties of chamomile flowers, which are due to the content of terpenoids, but also flavonoids and phenolic acids. This study aims to determine and compare the effects of the drying method on the metabolic profile of chamomile flowers from sustainable, organic practice. The flowers were dried using four different methods: in the sun at a temperature of around 30 °C for 4 days, in the shade at an average temperature of 20–25 °C for 7 days, in a dryer at a temperature of 105 °C for 24 h, and in a climate chamber at a temperature of 60 °C for 48 h. The drying method affects the color, aroma, dry biomass, and chemical profile of chamomile flowers. The biggest color change was between fresh chamomile flowers and chamomile flowers dried in a climate chamber at 105 °C for 24 h, and the smallest change was observed in flowers dried in the sun. The highest contents of polyphenolic compounds and antioxidant activity were measured in flower samples dried in the sun. Drying the flowers at 105 °C caused a significant decrease in total phenols and total flavonoids compared to the drying methods in the sun and shade. Drying at 60 °C for two days had the most significant negative effect on polyphenolic compounds. GC-MS analysis of chamomile essential oil revealed a total of 49 compounds. The most abundant compounds in all samples were α-bisabolol oxide A (19.6 to 24.3%), bisabolol oxide B (19.3 to 23.2%), and β-farnesene E (15.9 to 25.5%). β-Farnesene was identified in significantly lower amounts in sun-dried flowers compared to others, indicating its sensitivity to high light intensity. Volatile compounds spiroether Z, spiroether E, and matricarin were significantly reduced in samples dried at a temperature of 105 °C compared to others, which agrees with the aroma of dried flowers. Discrimination between samples based on chemical profiles showed similarity between samples dried in the sun and in the shade compared to samples dried at higher temperatures.

**Keywords:** antioxidant activity; chamomile; drying; essential oils; phenolic compounds; volatile compounds

## 1. Introduction

Chamomile (*Matricaria chamomilla* L.) is a plant from the Asteraceae family, genus Matricaria, characterized by five species widespread throughout Europe, central and southwestern Asia, northern Africa, western North America, and Macaronesia [1]. The stem of the plant is erect, hollow, and grows up to 60 cm in height. The leaves are located alternately, pinnately divided, and the flower heads grow individually on the stems. They are made up of bisexual yellow tubular flowers in the center and white petals on the

edge. They bloom from May until the beginning of autumn. They are aromatic and have a pleasant smell. The fruit is a light-brown achene.

Chamomile (*Matricaria chamomilla* L.) is a well-known annual plant that has been used worldwide for thousands of years [2] in the food industry, stomatology, medicine—including dermatology, gastroenterology, otolaryngology, pulmonology, internal medicine, radiotherapy, and pediatrics [3]—pharmaceutical, and cosmetic industries due to its proven beneficial properties, such as antimicrobial [4], anti-inflammatory [5], analgesic, sedative, and antiseptic properties [6], as well as anti-diarrheal, antioxidant [7], anticancer [8], and more. Chamomile possess benefits for human health and is traditionally used to treat ulcers, wounds, eczema, skin irritation, rash, gout, bruises, cracked nipples, mastitis, burns, ear and eye infections, neuralgia, conjunctivitis, rheumatic pain, hemorrhoids, smooth muscle relaxation, and other conditions [9–12]. The beneficial effects of dried chamomile flowers are also reflected in the relief of cramps and other inflammatory conditions of the digestive tract and menstrual cramps, and they can also be used as a mild sleep aid [13]. Most of the mentioned beneficial properties of chamomile flowers are due to the content of terpenoids and phenols [14], of which flavonoids and phenolic acids are the most important [2]. More than 120 secondary metabolites have been identified in chamomile flowers, including 36 flavonoids, 28 terpenoids, and others. Previously, it has been reported that chemical components, such as $\alpha$-bisabolol and cyclic ethers, have significant antimicrobial properties; umbelliferone is fungistatic, while chamazulene and $\alpha$-bisabolol are potent antiseptics [15]. Most of these compounds are available to the human body after the extraction process from plant tissues, e.g., extraction with solvents, hot water, alkalis, etc., but this may lead to degradation of the bioactive constituents. Methods for extraction with organic solvents, such as ethanol, acetone, and methanol, with different volume fractions of water are much more suitable for obtaining the higher yields of these chemical constituents. Apart from the type of farming system, the main characteristics of dried flowers are largely influenced by the drying conditions and methods [16]. Drying is one of the oldest methods of preserving food and plants, which aims to reduce the moisture content without affecting the quality of the raw material and can be performed by different methods, such as drying in the sun, in the shade, in conventional ovens, and in climate chambers; spray drying; and freeze drying, and each of them has its advantages and disadvantages [17]. The most widely used method for drying aromatic and medicinal herbs is shade or sun drying, but in this case, it is difficult to control factors such as temperature and humidity, and the possibility of contamination of the dried material is greater [18]. There are scientific papers in the literature dealing with the methods of drying chamomile flowers and dry extracts, and they continue to be researched to reliably investigate the efficacy of the different drying and extraction methods since the bioactive compounds contained in chamomile flowers are often sensitive and present in low concentrations [19–21]. Borsato et al. [22] found that chamomile flowers dried at 95 °C changed from an attractive yellow to an unattractive brown caramel color. Many authors have noted that the chemical composition of flowers and essential oil of aromatic plants varies depending on the drying method, thus recognizing the need for uniform results [23,24].

Agricultural systems that avoid the use of synthetic chemical pesticides, fertilizers, growth regulators, and other harmful chemicals are the basis of organic agriculture [25]. Organic agriculture preserves biological diversity, promotes soil fertility, and adapts production methods to tradition and locality, while growing diverse crops that tolerate growing conditions without chemicals. Sustainable agriculture, which can be described in many ways, includes ecological, biological, biodynamic, bioecological, regenerative, organic, conservation agriculture [26]. It can also be defined as an agricultural practice that does not affect the chemical pollution of agricultural resources, such as soil and water, increasing and maintaining biodiversity and ensuring food safety. Sustainable cultivation has an impact on environmental protection and promotes rural development [27]. Chamomile (*Matricaria chamomilla* L.) is an extremely grateful plant for cultivation according to ecological principles and sustainable agricultural practices, whose principles we adhered to

during research. Modern technological methods, which include climate chambers and various ovens, and traditional drying methods, such as sun and shade drying, can be counted among sustainable, green drying methods because they do not use non-renewable energy sources, but continue to have different effects on the extraction of secondary plant compounds, especially polyphenolic compounds. In addition to the drying method, the extraction of chemical compounds in essential oils is also affected by the type of solvent. Therefore, this study aims to determine and compare the effects of the drying method on the metabolic profile of chamomile (*Matricaria chamomilla* L.) flowers from sustainable, organic seeds and cultivation in the region of Slavonia, the eastern part of Croatia.

## 2. Materials and Methods

### 2.1. Plant Growth and Drying Methodology

The study was conducted in Brod-Posavina County, Slavonia region, in the eastern part of Croatia, about 8 km west of Slavonski Brod, on the experimental field "Slobodnica" of the University of Slavonski Brod (45°09′58.5″ N and 17°56′52.9″ E) at 87 m altitude. The climate of Brod-Posavina County is a temperate continental climate with warm summers and moderately cold winters. The experimental fields are located in an area where the monthly temperature is above 10 °C for more than four months, the average temperature in the warmest month is below 22 °C, and the average annual precipitation is 700–800 mm. The area is characterized by winter temperatures that can fall below 0 °C and summer temperatures that can rise above 30 °C [28] and average annual humidity of 72%. The area along the Sava River is predominantly characterized by alluvial amphigley soils that are moistened by soil and surface water. The experimental fields are located in an area where excessive wetting by surface water occasionally occurs—pseudogley [29].

Organically grown seeds of *Matricaria chamomilla* L. were purchased from domestic organic production by the company Espresso d.o.o. from Lužani, Brod-Posavina County. The seeds were sown by hand on 28 October 2021, on a 0.6-hectare experimental field, and the sowing rate was 8 kg per ha. Seeds started to germinate after 7 days, and the plant density was 34 plant/m². An organic cultivation system was used, and the flowers (Figure 1) were harvested by hand on 15 May 2022 and immediately taken to the laboratory of the Biotechnical Department of the University of Slavonski Brod, where the samples were dried using various drying methods.

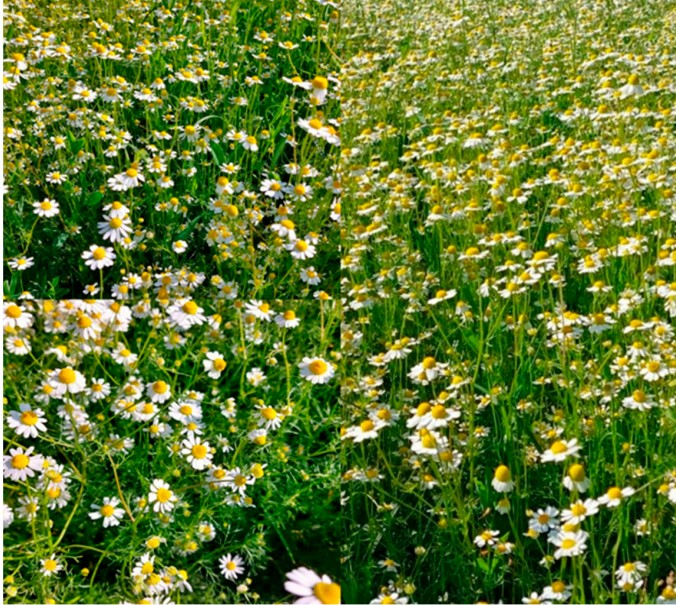

**Figure 1.** Chamomile plants in the flowering stage in the experimental field of the University of Slavonski Brod, Croatia.

The flower samples consisted of 250 g in four replicates and were randomly selected for each drying method. Four types of drying of the plant material were performed:

- drying in the sun (temperature around 30 °C for 4 days)—SUN;
- drying in the shade (average temperature of 20–25 °C for 7 days)—SH;
- drying in the dryer (Memmert, model UNE 200) (temperature of 105 °C for 24 h)—D;
- drying in the climatic chamber (Memmert, model ICH L eco) (temperature of 60 °C for 48 h)—KK60.

The average relative humidity during the drying of chamomile flowers was 64%, while the average temperature was 18.7 °C, according to the state hydrometeorological institute. Dried samples were used for further analysis.

### 2.2. Polyphenols and Antioxidant Activity Measurements

For the measurement of polyphenolic compounds and antioxidant activities, 60 mg of dried tissue was extracted in 2 mL of 80% methanol. Tissue homogenization was performed by a Mixer Mill MM 400 (Retsch, Haan, Germany) for 2 min at 30 Hz, after which the extracts were mixed in a tube rotator for 30 min at 15 rpm. The extracts were then centrifuged for 10 min at 13,000 rpm (Eppendorf centrifuge), and the supernatants were used for spectroscopy measurements on a spectrofluorometer microplate reader (Tecan Infinite M200, Männedorf, Switzerland). Antioxidative activities were measured by using a UV/VIS spectrophotometer (BioSpec-1601 E, Shimadzu, Kyoto, Japan).

Total phenolics (TP) were determined following the Folin–Ciocalteu method, as reported by Singleton and Rossi [30]. TP data were expressed as equivalents of gallic acid per dry weight (mg GAE $g^{-1}$ dw). Total phenolic acids (TPA) were measured using Arnow's reagent, according to the European Pharmacopoeia (2004) [31], and the results were presented as equivalents of caffeic acid per dry weight (mg CAE $g^{-1}$ dw). Total flavonoids (TFL) were measured using the $AlCl_3$ method [32], and results were presented as equivalents of catechin per dry weight (mg CE $g^{-1}$ dw). Antioxidant activity was measured by the DPPH radical scavenging capacity assay, according to Brand-Williams et al. [33]. The obtained results of the DPPH assay were expressed as µmol Trolox equivalents per gram dry weight (µmol TE $g^{-1}$ dw).

### 2.3. Gas Chromatography–Mass Spectrometry (GC/MS) Analysis of Essential Oils and n-Hexane Chamomile Extracts

Essential oils were obtained by 1 h hydrodistillation in a Clevenger apparatus of 8–10 g of homogenized dried plant material. In detail, finely powdered plant material was placed in a round-bottom flask, and 500 mL of water was added. Heat was applied to the flask, causing the water to boil and produce steam, which carried the volatile essential oils along with it. After 1 h, the essential oil was then separated from the aqueous layer and transferred into the vial and kept at 4 °C until analysis [34].

For the preparation of chamomile extracts in *n*-hexane, homogenized dried plant material (100 mg) was extracted with 1800 µL *n*-hexane, containing 0.001% *n*-tridecane as internal standard, and then sonicated for 10 min in an ultrasonic bath. After centrifugation at 14,500× *g*, 1000 µL of supernatant was transferred into a new vial for analysis. Each extract was isolated in triplicate.

Volatile constituents were determined by GC/MS using an Agilent 7890A gas chromatograph fitted with a fused silica HP-5MS UI capillary column (30 m × 0.25 mm, 0.25 µm film thickness), coupled to an Agilent 5975C mass selective detector.

GC/MS analysis and the identification of the volatile constituents were conducted following the method reported by Cavar Zeljkovic et al. [35], but with some modifications in the operating conditions as follows: inlet pressure 9.35 psi, injector temperature 250 °C, detector temperature 280 °C, and split ratio for essential oils 1:9.

### 2.4. Statistics

Statistical analysis was performed in RStudio (R Software version 4.1.0) using packages compositions, agricolae, ggplot2, pls.

Multiple one-way ANOVAs followed by Duncan's tests for multiple comparisons (on data in logit-scale) were performed to evaluate the impact of drying on the essential oils and the hexane extracts. The effect of drying was further examined and visualized via compositional PLS-DA biplots (taking the type of drying as the response and expressing the essential oils and hexane extracts compositions in clr coefficients).

## 3. Results and Discussion

### 3.1. Dried Sample Characteristics

The dried samples differed visually in color, depending on the drying methods used. As can be seen in Figure 2, the samples dried in the sun (SUN) and shade (SH) were brightly colored and resembled the fresh material. Their aromas were easily recognizable and intense. The samples dried at 105 °C (D) and 60 °C (KK60) lost their natural color and became brownish, especially the D samples. D samples also lost their characteristic chamomile aroma, while KK60 samples were still recognizable, although the intensity of the aroma was lower compared to air-dried samples. Borsato et al. (2005) [22] previously reported that chamomile flowers dried at 95 °C changed from an attractive yellow to an unattractive brown caramel color. We noticed that when drying at higher temperatures, the chamomile flower changed color from a pleasant yellow (SUN) to an unattractive brown (D) (Figure 2).

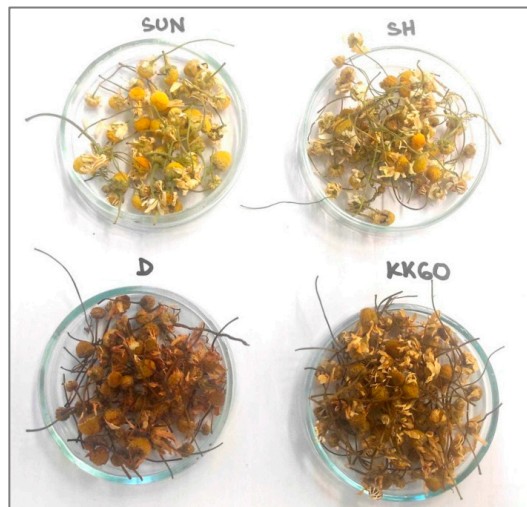

**Figure 2.** Samples of chamomile flowers dried in the sun for 4 days (SUN), in shade for 7 days (SH), in drier at 105 °C for 24 h (D), and in climatic chamber at 60 °C for 2 days (KK60).

Chamomile flowers were dried to their constant weight by various drying methods. The size of the sample was 250 g. After drying, the weight of chamomile flowers ranged from 42.03 g (D) to 56.97 g (SUN). The average weight of four repetitions in the SUN treatment was 54.95 g, the SH treatment was 52.09 g, the D treatment was 44.27 g, and the KK60 treatment was 52.42 g (Figure 3).

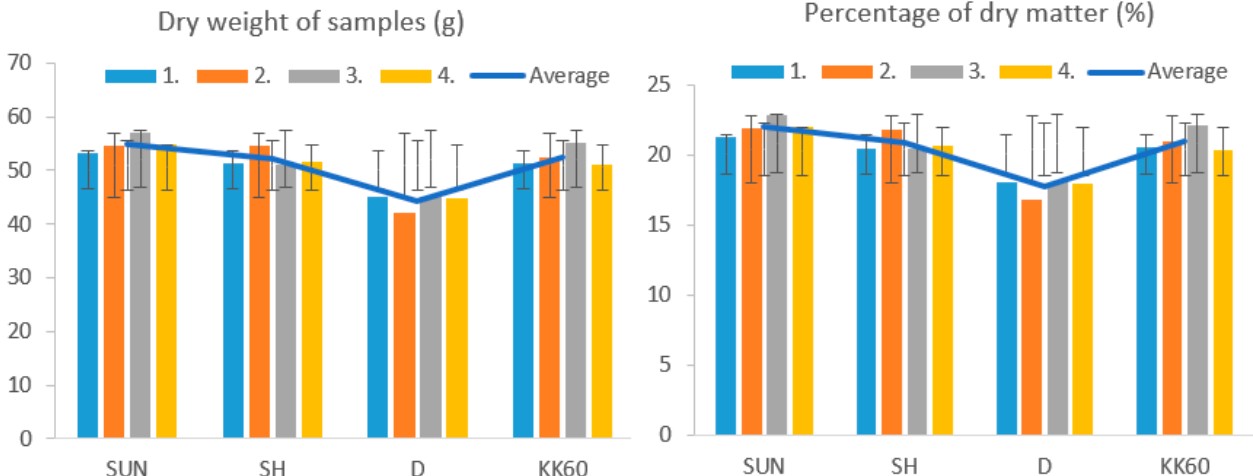

**Figure 3.** Dry weight and percentage of dry matter of flower samples (g) after sun drying for 4 days (SUN), shade drying for 7 days (SH), drying in the dryer at 105 °C for 24 h (D), and drying in the climatic chamber at 60 °C for 48 h (KK60). Values are an average of 4 replicates ± SD.

Dried samples had an average moisture content between 16.81 and 22.79%. The lowest moisture content of 16.81% was observed in the D treatment and the highest moisture content of 22.79% in the SUN treatment. The average moisture content of four repetitions in the SUN treatment was 21.98%, in the SH treatment was 20.84%, in the D treatment was 17.71% g, and in the KK60 treatment was 20.96% (Figure 3). Similar results were reported by Abbas et al. (2021) [19]. It is important to note that this moisture content in the samples is not suitable for storing and selling the product.

### 3.2. Polyphenols and Antioxidant Activity

Total polyphenols (TP), total flavonoids (TFL), total phenolic acids (TPA), and antioxidant activity measured by the DPPH assay were evaluated using spectroscopic methods and are presented in Figure 4. As can be seen, the highest contents of polyphenolic compounds and antioxidant activity were measured in flower samples dried in the air exposed to the sun (SUN); TP content was 32.02 mg GAE $g^{-1}$ dw, TFL content was 12.36 mg CE $g^{-1}$ dw, TPA content was 15.14 mg CAE $g^{-1}$ dw, and antioxidant activity was 0.66 mg TE $g^{-1}$ dw. Drying the flowers in the shade (SH) did not influence significantly TP, TFL, TPA, or antioxidant activity, compared to SUN samples. Short drying of flowers at 105 °C caused a significant decrease of TP and TFL (23.62 mg GAE $g^{-1}$ dw and 7.51 mg CE $g^{-1}$ dw, respectively) compared to the SUN and SH drying methods. There was a trend of a slight decrease of TPA and antioxidant activity in flowers dried at 105 °C compared to air-dried samples (SUN and SH), although it was not statistically significant. The most prominent negative effect on polyphenolic compounds in chamomile flowers was caused by drying at 60 °C for two days (KK60); TP, TFL, and TPA contents were decreased by 41.3%, 65.2%, and 44.2%, respectively, and antioxidant activity was reduced 39.4%, compared to SUN samples. Different methods of drying chamomile flowers and their influence on the content of chemical components have been investigated by other authors. Harbourne et al. (2009) [36] determined a statistically significant decrease in the phenol concentration in samples dried at higher temperatures (80 °C). Based on our results, short drying at 105 °C caused fewer negative changes in polyphenol content and antioxidative activity compared to the prolonged drying at 60 °C.

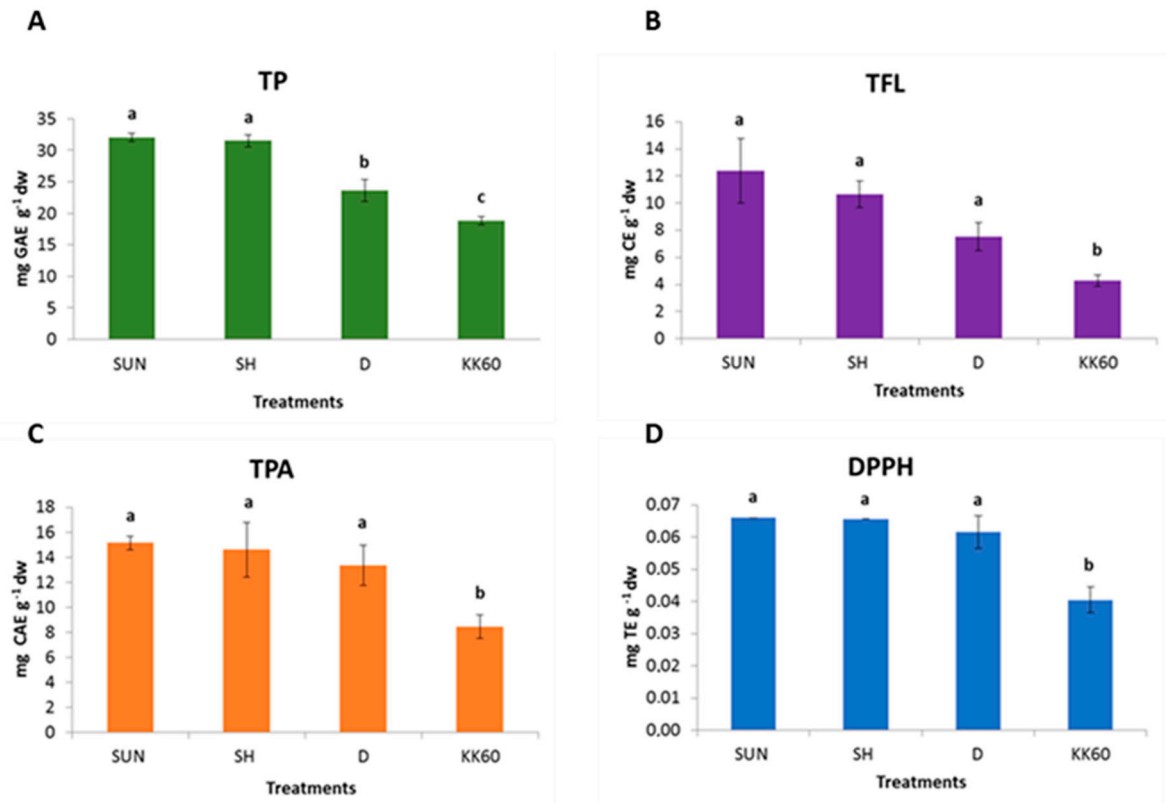

**Figure 4.** Polyphenolic compounds: (**A**) total polyphenols (TP), (**B**) total flavonoids (TFL), (**C**) total phenolic acids (TPA), and (**D**) antioxidant activity measured by DPPH assay in chamomile flowers dried in air exposed to direct sun for 4 days (SUN), dried in air exposed to shade (SH), dried in the drier at 105 °C for 24 h (D), and dried in the climate chamber at 60 °C for 2 days (KK60). Data are average, n = 4 ± SD. Different letters denote significant differences in compound level between treatments ($p < 0.05$, one-way ANOVA, Duncan's test).

### 3.3. Essential Oils

GC-MS analysis of chamomile essential oil identified 49 compounds in total (Figure 5). Salamon et al. (2023) [37] recorded between 23 and 43 chemical components in the essential oil extracted from chamomile flowers in different sites in Albania, and Aćimović et al. (2021) [38] between 47 and 57 chemical components in essential oil. The most abundant compounds in all samples are α-bisabolol oxide A (ABOLA) (19.6 to 24.3%), bisabolol oxide B (BIOB) (19.3 to 23.2%), and β-farnesene E (BFAR) (15.9 to 25.5%). These several essential oils chemotypes are known as antimicrobial, spasmolytic, antiepileptic, mitogenic, and insecticidal mediums [39]. The highest level of ABOLA was found in KK-60, then SUN, D, and SH, while the highest level of BIOB was found in the SUN sample, then in D, SH, and KK-60. Hajaji et al. (2018) [40] obtained results that showed that α-bisabolol (ABOL) can activate the programmed process of cell death in the promastigote stage of the parasite. BFAR was the highest in flowers dried at high temperatures for 24 h, while it was the lowest in samples exposed to the sun. Chamazulene (CHAM) was the highest in sun-dried samples (SUN sample, 4.3%), and it was significantly decreased upon high-temperature drying (D sample, 1.5%). Chamazulene is a well-known constituent of chamomile extracts and may be in part responsible for the chamomile aroma, which was well preserved in sun- and shade-dried samples compared to other drying treatments. Ghasemi et al. (2016) [41] determined that the content of chamazulene and α-bisabolol in chamomile oil is influenced by environmental conditions and the genetic background. We demonstrated that their content is also influenced by drying methods, which is in agreement with data obtained by Abbas et al. (2021) [19]. On the other hand, GMUU was low in air-dried samples, SUN and SH (2.2 and 2%, respectively), and significantly more abundant in the D sample (4.1%).

Low-abundant compounds were grouped, and they contributed 5.1 to 7.1% of the mixture. The terpenes composition of the essential oil is dependent on the stage of maturity and the cultivation system [42], as well as the chamomile cultivars [43].

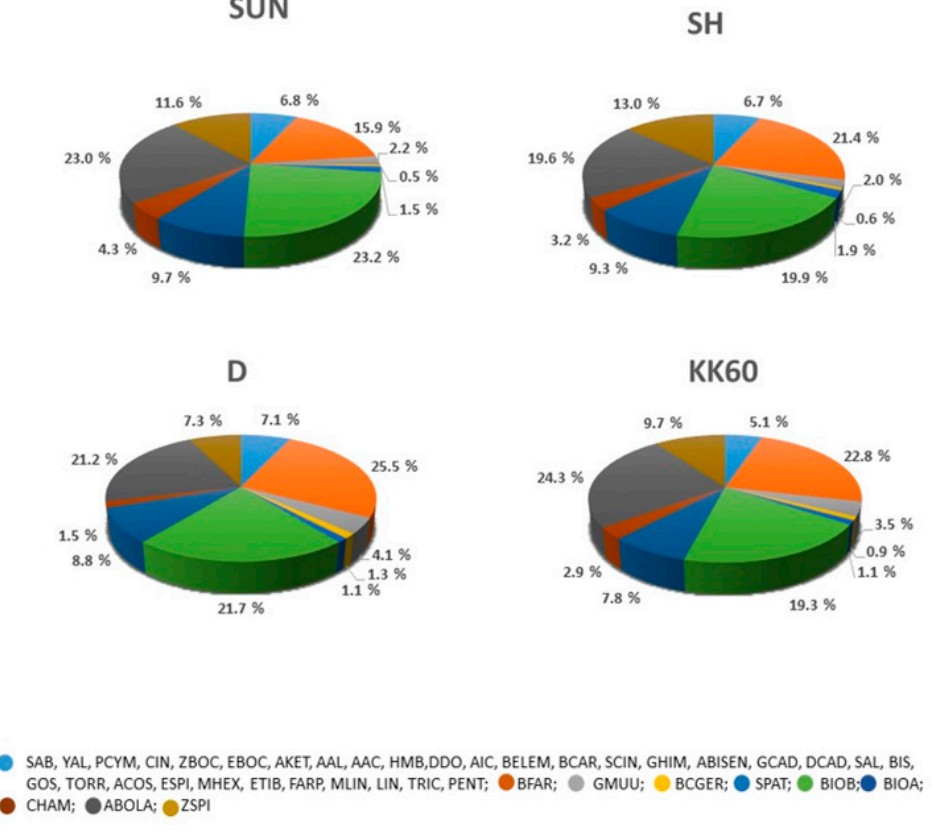

**Figure 5.** Abundance of essential oil compounds in chamomile flowers dried in air exposed to direct sun for 4 days (SUN), in air exposed to shade (SH), dried in a drier at 105 °C for 24 h (D), and dried in a climate chamber at 60 °C for 2 days (KK60). Bicyclogermacrene (BCGER), Bisaboladien-4-ol-2,7Z (BIS), α-Bisabolene Z (ABISEN), α-Bisabolol (ABOL), α-Bisabolol oxide A (ABOLA), epi-α-Bisabolol (EPIAB), Bisabolone oxide A (BIOA), Bisabolol oxide B (BIOB), γ-Cadinene (GCAD), δ-Cadinene (DCAD), Chamazulene (CHAM), α-Costol (ACOS), Dauca-5,8-diene (DAU), β-Farnesene E (BFAR), (2E,6E)-Farnesyl pentanoate (FARP), Farnesol 2Z,6Z (FAR), Gossonorol (GOS), γ-Himachalene (GHIM), Linoleic acid (LIN), Methyl linoleate (MLIN), α-Muurolene (AMUU), γ-Muurolene (GMUU), Methyl hexadecanoate (MHEX), Pentacosane (PENT), Salvial-4(14)-en-1-one (SAL), dehidro-Sesquicineole (SCIN), Spathulenol (SPAT), Spiroether Z (ZSPI), Spiroether E (ESPI), (E)-Tibetin spiroether (ETIB), *n*-Tricosane (TRIC), (E)-Tibetin spiroether (ETIB), and Torreyol (TORR) (See Table S1).

*3.4. Volatile Compounds Extracted by n-Hexane*

GC-MS analysis identified 15 volatile compounds extracted by *n*-hexane (Figure 6).

Some of the compounds have been already identified in the essential oil reported above, like ABOLA, BFAR, ZSPI, etc. The drying methodology affected the level of certain compounds. The most abundant volatile compound found in chamomile flowers was Spiroether Z (ZSPI). Similar results were obtained by Aćimović et al. (2021) [38]. ZSPI did not change significantly in SUN, SH, and KK60 samples (34–36%), but decreased significantly after flower drying at high temperature (D sample, to 28%). Spiroether E (ESPI) and Matricarin (MATR) were also significantly decreased in D samples compared to SUN, SH, and KK60. The amounts of volatile compounds agree with the aroma of dried samples. Spiroether E and Spiroether Z from the essential oil of chamomile showed specific repression in the production of alphatoxin AFG (1) by Aspergillus parasiticus [44]. On the other hand, Tridecane (TRID) and Pentacosane (PENT) were significantly increased

in flowers dried at high temperatures compared to other samples. β-Farnesene E (BFAR) was identified in significantly lower amounts in SUN-dried flowers compared to others, suggesting its sensitivity to high light intensity. BFAR compound is a pheromone for some insects and may have importance in some forms of ecological pest control [37]. α-Bisabolol oxide A (ABOLA) and α-Bisabolol oxide B (ABOLB) were highest in samples dried at higher temperatures: SUN samples and SUN and D samples, respectively. Bicyclogermacrene (BCGER), Spathulenol (SPAT), α-Bisabolol (ABOL), (E)-Tibetin spiroether (ETIB), and *n*-Tricosane (TRIC) were identified in low amounts in all samples (below 1% in mixture). α-Bisabolol was the highest in the SUN treatment, but without statistically significant differences compared to others.

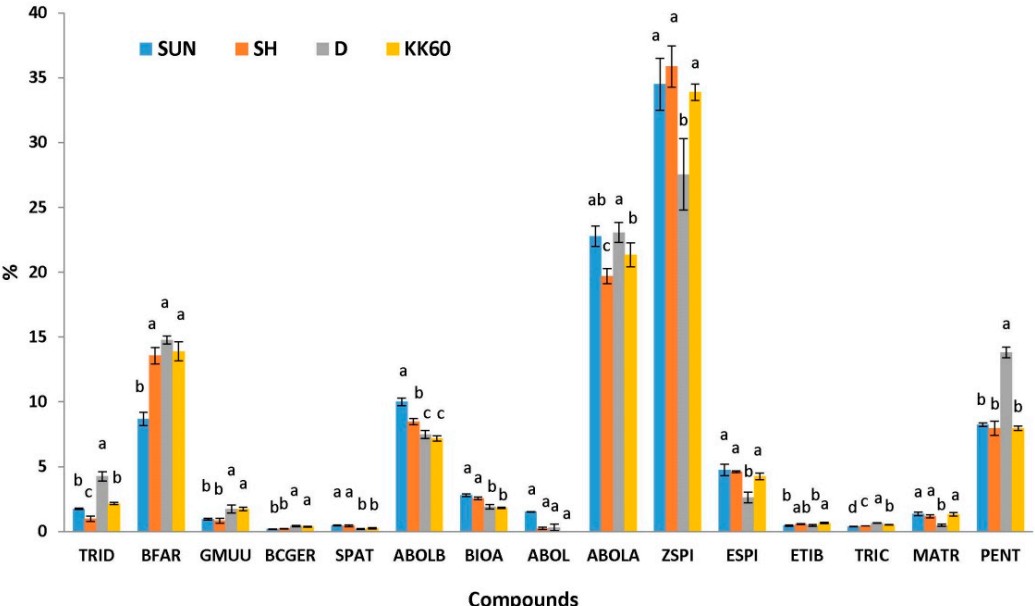

**Figure 6.** Volatile compounds extracted by *n*-hexane and identified by GC-MS and presented as % in mixture: Tridecane (TRID), β-Farnesene E (BFAR), γ-Muurolene (GMUU), Bicyclogermacrene (BCGER), Spathulenol (SPAT), α-Bisabolol (ABOL), α-Bisabolol oxide A (ABOLA), α-Bisabolol oxide B (ABOLB), α-Bisabolone oxide A (BIOA), Spiroether Z (ZSPI), Spiroether E (ESPI), (E)-Tibetin spiroether (ETIB), *n*-Tricosane (TRIC), Matricarin (MATR), Pentacosane (PENT). Data are average, n = 4 ± SD. Different letters denote significant differences in compound level between treatments (*p* < 0.05, one-way ANOVA, Duncan's test).

### 3.5. Relationship between Drying Methods and Metabolomics Profile

Our findings show that the drying methodology influences the metabolomics profile of chamomile flowers. PLS-DA biplots (Figures 7 and 8) provide a closer look at these relationships. The effect of drying technology on polyphenolic compounds and antioxidant activity is visualized in the PLS-DA biplot shown in Figure 7. The first two PLS components capture 97.92% of the variability in the analyzed traits (the vast majority, 93.76%, being explained by the first PLS component). It can be seen that higher content of polyphenolic and antioxidant activity was measured in the SUN and SH samples, while lower content was observed in the KK60 samples. Azizi et al. (2009) [45] determined that the maximum content of chamomile essential oil was obtained when drying at lower temperatures and in shaded areas.

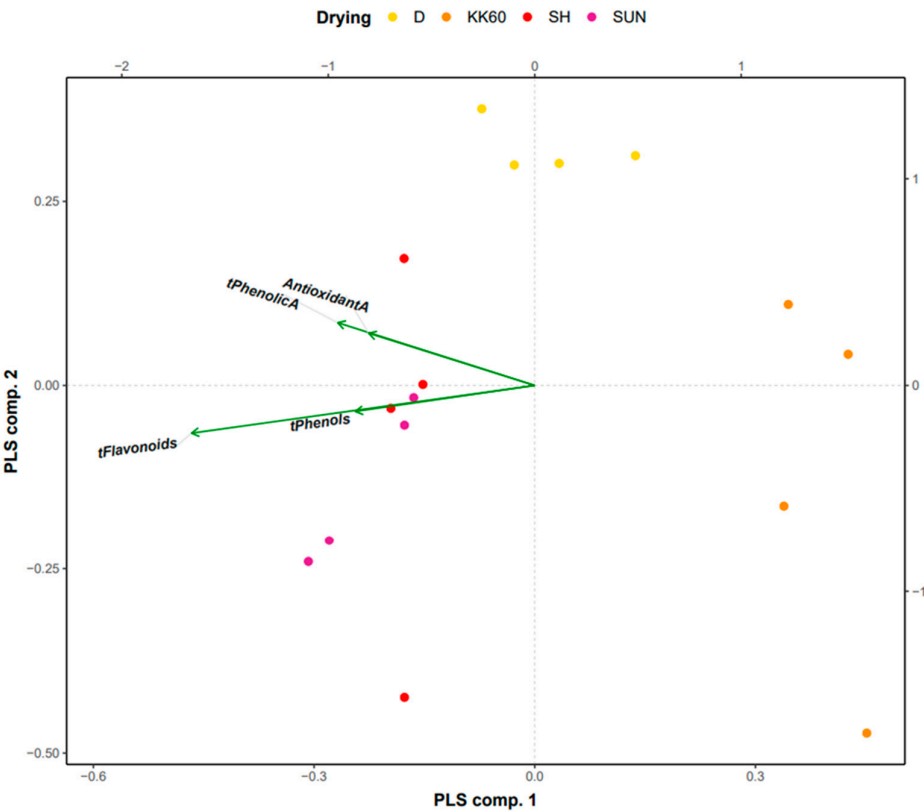

**Figure 7.** PLS-DA biplot showing the relationship between drying methodology and polyphenolic compounds, as well as antioxidant activity.

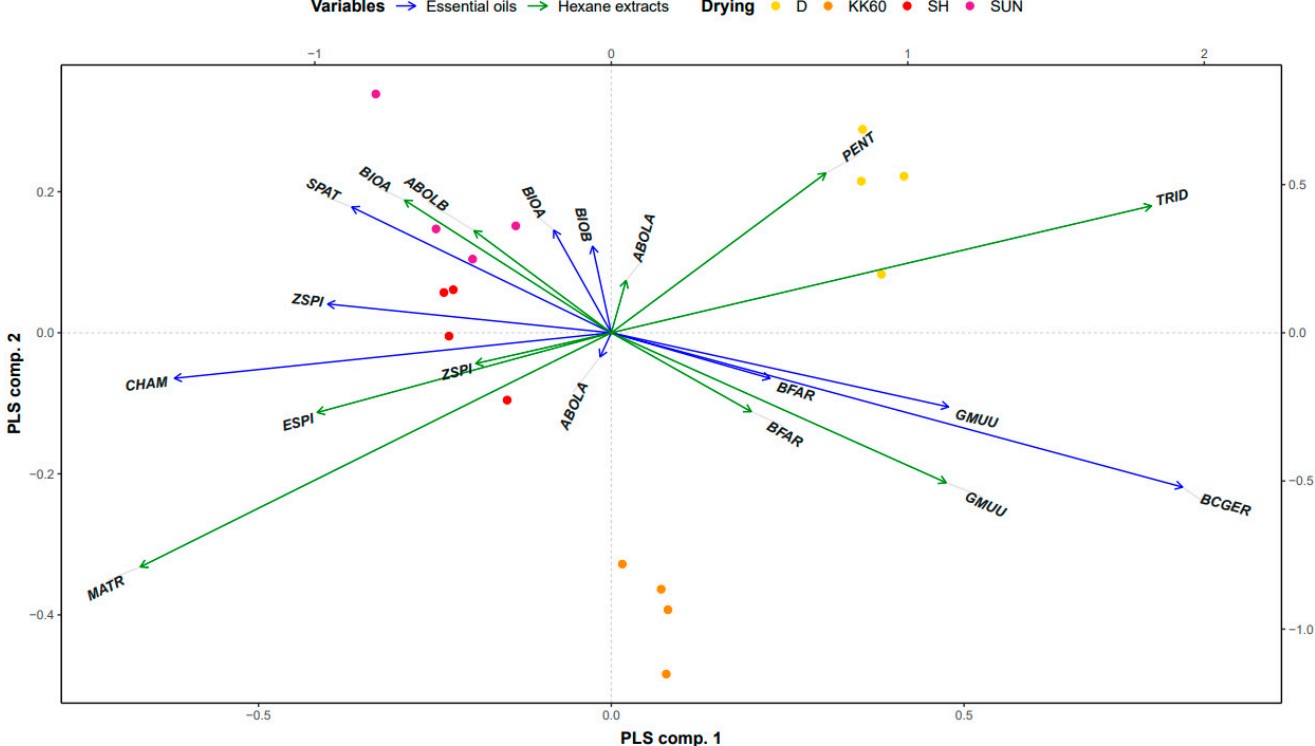

**Figure 8.** PLS-DA biplot showing the relationship between drying methodology and polyphenolic compounds and essential oils and *n*-hexane compounds.

The impact of the drying method on GC-MS-analyzed compounds (essential oils and *n*-hexane compounds) is visualized in the PLS-DA biplot shown in Figure 8. The first two PLS components capture 81.36% of the variability in the considered compounds (the majority, 71.41%, being explained by the first PLS component). The SUN and SH samples showed similar metabolomic profiles, while the D samples differed from them more than the KK60 samples. Compounds with relatively high (resp. low) levels in the SUN and SH samples are located on the left (resp. on the right). The D samples were characterized especially by a relatively high abundance of TRID and PENT and contrarily by a relatively low abundance of MATR, CHAM, ESPI, and ZSPI.

## 4. Conclusions

Based on the obtained results, we can conclude that the drying method affects the color, aroma, and weight of dry flowers, as well as the quality and composition of the essential chamomile oil. The biggest color change was between fresh chamomile flowers and chamomile flowers dried in a climate chamber at 105 °C for 24 h, followed by samples dried in a dryer at 60 °C for 48 h, and the smallest change was observed in flowers dried in the sun and flowers dried in the shade. The findings suggested that the contents of total phenols, total flavonoids, total phenolic acids, and antioxidant activity were higher in flower samples dried in the sun and in the shade compare to at higher temperatures.

Sun- and shade-dried samples were more similar in essential oils and volatile compound metabolic profiles than samples dried at higher temperatures. The volatile compounds spiroether Z (ZSPI), spiroether E (ESPI), and matricarin (MATR) were similarly abundant in the sun, in the shade, and drying in the climatic chamber at 60 °C for 48 h, but significantly decreased in drying in the dryer at 105 °C for 24 h.

Different methods of drying can cause a decrease and/or increase in some compounds, which may be desirable in certain phytomedical situations, so it is important to know and apply different methods after harvesting when drying chamomile flowers.

**Supplementary Materials:** The following supporting information can be downloaded at: https://www.mdpi.com/article/10.3390/su152115373/s1, Table S1: Essential oil compounds in chamomile flowers dried on air exposed to direct sun for 4 days (SUN), on air exposed to shadow (SH), dried in a drier at 105 °C for 24 h (D), and drying in the climate chamber.

**Author Contributions:** Conceptualization, T.B.-L.; methodology, T.B.-L., R.B., K.M., I.O., B.S.-S., S.Ć.Z., N.Š. and P.T.; validation, S.Ć.Z. and B.S.-S.; formal analysis, I.O., S.Ć.Z., N.Š. and R.B.; investigation, T.B.-L., R.B., K.M., I.O. and B.S.-S.; resources, T.B.-L., K.M., B.S.-S. and P.T.; data curation, T.B.-L., S.Ć.Z., P.T. and B.S.-S.; writing—original draft preparation, T.B.-L. and B.S.-S.; writing—review and editing, K.M., R.B., I.O., S.Ć.Z., N.Š. and P.T.; visualization, R.B., I.O. and N.Š.; supervision, K.M and S.Ć.Z.; project administration, T.B.-L. All authors have read and agreed to the published version of the manuscript.

**Funding:** University of Slavonski Brod and Brod-Posavina County (project: The influence of different methods of soil tillage and fertilization on the yield of field crops). This research was also funded by the project no. MZE-RO0423 by the Ministry of Agriculture, Czech Republic.

**Institutional Review Board Statement:** Not applicable.

**Informed Consent Statement:** Not applicable.

**Data Availability Statement:** Not applicable.

**Conflicts of Interest:** The authors declare no conflict of interest.

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
