# Peer review of "Effect of Drying Methods on Chemical Profile of Chamomile (Matricaria chamomilla L.) Flowers"

_sustainability, doi:10.3390/su152115373_

Round 1

Reviewer 1 Report

Dear Authors,

I had the opportunity to review your article. The study is, to my knowledge, one of the most comprehensive of its kind. We consider  this study aims to determine 86 and compare the effects of the drying method on the metabolic profile of chamomile (Mat- 87 ricaria chamomilla L.) flowers from organic seeds and cultivation in the region of Slavonia, 88 the eastern part of Croatia. For this reason, the study is an original study with a high potential to contribute to the related literature and the journal with its potential citation numbers. However, I have some concerns that need to be clarified/corrected via the draft. Please find them according to the list below:

1.       Please double-check the abstract and conclusion section, and could be developable.

2.       The language needs a minor revision, please double-check since it requires critical revisions/corrections in its syntax, punctuation and grammar.

3.       To present the abbreviation; you must first give a full name, then an abbreviation. Please make standard for all texts in the text.

4.       Line 2: Title of article please correct perhaps "Effect of drying methodology on the chemical profile of chamomile (Matricaria chamomilla L.) flowers” ???????

5.       Please, The brand and model of the device you use in the drying process should be written. In addition, during this measurement, humidity ratios in drying environments should also be taken and if so, it should be written. It is known to be an important factor in the drying process.?????????????

6.       The information given in the introduction could be checked again. The order given must be adjusted.

7.       Please, A little more information on the taxonomy and systematics of chamomile (Matricaria chamomilla L.) can be given.

8.       Line 333: Please check to the “TableS1” I did not see a table in the article. Also volatile compounds and essential oil compounds should be given in a table. For a clearer understanding.!!!!!!!!

9.       The discussion section could be further expanded and a few additions could be made.

10.    Literature should be increased, especially in the introduction, and please double-check.

11.    Please, double-check conlusions; should be more clearly.????

Overall, the present study is an original study. So, it can be considered as a potential work of its kind.

Regards

Author Response

I had the opportunity to review your article. The study is, to my knowledge, one of the most comprehensive of its kind. We consider this study aims to determine 86 and compare the effects of the drying method on the metabolic profile of chamomile (Mat- 87 ricaria chamomilla L.) flowers from organic seeds and cultivation in the region of Slavonia, 88 the eastern part of Croatia. For this reason, the study is an original study with a high potential to contribute to the related literature and the journal with its potential citation numbers. However, I have some concerns that need to be clarified/corrected via the draft. Please find them according to the list below:

Reviewer 1

Comment to Authors

Answer

Please double-check the abstract and conclusion section, and could be developable.

Corrected as suggested.

The conclusion and abstract is developed.

The language needs a minor revision, please double-check since it requires critical revisions/corrections in its syntax, punctuation and grammar.

Language has been checked and revised.

To present the abbreviation; you must first give a full name, then an abbreviation. Please make standard for all texts in the text.

Corrected as suggested.

Line 2: Title of article please correct perhaps "Effect of drying methodology on the chemical profile of chamomile (Matricaria chamomilla L.) flowers” ???????

Corrected as suggested.

Please, The brand and model of the device you use in the drying process should be written. In addition, during this measurement, humidity ratios in drying environments should also be taken and if so, it should be written. It is known to be an important factor in the drying process.?????????????

Corrected as suggested.

The brand and model of the device are recorded.

The requested information has been added to the manuscript.

The information given in the introduction could be checked again. The order given must be adjusted.

Corrected as suggested.

We adjusted the order and give a little more information on taxonomy and systematics of chamomile.

Please, A little more information on the taxonomy and systematics of chamomile (Matricaria chamomilla L.) can be given.

More information  on taxonomy and systematics are added.

Line 333: Please check to the “TableS1” I did not see a table in the article. Also, volatile compounds and essential oil compounds should be given in a table. For a clearer understanding.!!!!!!!!

Corrected as suggested.

Table S1. ” Essential oil compounds in chamomile flowers dried on air exposed to direct sun for 4 days (SUN), on air exposed to shadow (SH), dried in a drier at 105°C for 24 hours (D), and drying in the climate chamber“ nserted on the end of article.

The discussion section could be further expanded and a few additions could be made.

Corrected as suggested.

Results and Discussion section is expanded and few additions are made.

Literature should be increased, especially in the introduction, and please double-check.

Corrected as suggested.

Literature is increased and checked.

Please, double-check conlusions; should be more clearly.????

Corrected as suggested.

The conclusion is check, developed, and we hope more clearly now.

Reviewer 2 Report

line_178:   It is essential to present the chamomile water loss curve throughout the process.
line_100:
  it is necessary to inform the average annual relative humidity. 
line_117 and line_118: it is necessary to inform relative humidity of air for 4 days.
line_119 line_121: what is the air flow rate of drying air?
line_188: You cannot compare your data to Borsato (2005) because he used different drying methods.
line_213: moisture content between 16.81-22.79% w.b.  These moisture contents are not suitable for storage and marketing of the product. It was not possible to reach smaller contents due to the drying methods used.
line_386: inappropriate title because there is nothing in your article that can be referred to as drying technology. Only drying methods.
line_440: only five references on drying out of thirty-nine cited.

Author Response

Reviewer 2

Comment to Authors

Answers

line_178:   It is essential to present the chamomile water loss curve throughout the process.

The curve of water loss from the chamomile flower was not measured because we felt that for this research it was not necessary because the work emphasizes polyphenols and chemical substances of flower extraction. In the paper, more emphasis is placed on the quality of the essential oil and not so much on the drying technology.

line_100:  it is necessary to inform the average annual relative humidity. 

Corrected as suggested.

Information added into the article.

line_117 and line_118: it is necessary to inform relative humidity of air for 4 days.

Corrected as suggested.

Information added into the article.

line_119 line_121: what is the air flow rate of drying air?

The manufacturer for dryer does not specify in the operating instructions what the speed and volume of the air flow is for each 10% of the possibility of adjusting the number of revolutions of the fan.

line_188: You cannot compare your data to Borsato (2005) because he used different drying methods.

Corrected as suggested.

Now we only list the literary source and do not compare.

line_213: moisture content between 16.81-22.79% w.b.  These moisture contents are not suitable for storage and marketing of the product. It was not possible to reach smaller contents due to the drying methods used.

Corrected as suggested.

We fully agree with the comment, we added a sentence that explains it, but this product was not intended for storage and sale of products.

line_386: inappropriate title because there is nothing in your article that can be referred to as drying technology. Only drying methods.

Corrected as suggested.

Title: “Effect of drying methods on chemical profile of chamomile (Matricaria chamomilla L.) flowers”

line_440: only five references on drying out of thirty-nine cited.

Corrected as suggested.

We added 2 more references on drying metode.

Reviewer 3 Report

I have put some comments at the same pdf file.

My main suggestions:

Design: I missed plant density and germination rate data.

Methodology: I think distilaltion should be better described (plant/eater rate (Citation).

Results: It is very important to know the extraction yield. We do not have this information during all the manuscript.

No issues detected.

Author Response

Reviewer 3

Comment to Authors

Answers

Design: I missed plant density and germination rate data.

Corrected as suggested.

Requested data have been added to material and methods.

Methodology: I think distilaltion should be better described 

Corrected as suggested.

Details on distillation are added to material and methods, subtitle 2.3.

Essential oils were obtained by 1 h hydrodistillation in Clevenger apparatus of 8-10 g of homogenized dried plant material. In detail, finely powdered plant material is placed in a round-bottom flask, and 500 mL of water is added. Heat is applied to the flask, causing the water to boil and produce steam, which carries the volatile essential oils along with it. After 1 h, the essential oil is then separated from the aqueous layer, and transferred into the vial and kept at 4 °C until analysis (Fagbemi et al. 2021).

REF: Fagbemi KO, Aina DA, Olajuyigbe OO. Soxhlet Extraction versus Hydrodistillation Using the Clevenger Apparatus: A Comparative Study on the Extraction of a Volatile Compound from Tamarindus indica Seeds. ScientificWorldJournal. 2021 Dec 2;2021:5961586. doi: 10.1155/2021/5961586.

It is very important to know the extraction yield. We do not have this information during all the manuscript.

We agree that the essential oils yield is highly important from both producers and processors in the industrial practice, but here, in this study, we focused on the quality, but not quantity of the essential oils (and phenolic extracts) in Chamomile.

Round 2

Reviewer 2 Report

Despite your answer about line 178, it should be noted that, in the study on drying, it is important to present the drying curve, so that you can observe this curve the drying rate, which is the parameter that determines the final quality of the oil. The drying rate is determined by the temperature and speed of the drying air. And the methods you've used lack that detail. That's why it was necessary to change the title.